# Genome-wide siRNA screens identify RBBP9 function as a potential target in Fanconi anaemia-deficient head-and-neck squamous cell carcinoma

Govind Pai [1][✉], Khashayar Roohollahi[1], Davy Rockx[1], Yvonne de Jong[1], Chantal Stoepker[1], Charlotte Pennings[1], Martin Rooimans[1], Lianne Vriend[1], Sander Piersma[2], Connie R. Jimenez[2], Renee X. De Menezes [3,7], Victor W. Van Beusechem[4], Ruud H. Brakenhoff[5], Hein Te Riele[6], Rob M. F. Wolthuis[1] & Josephine C. Dorsman [1][✉]

Fanconi anaemia (FA) is a rare chromosomal-instability syndrome caused by mutations of any of the 22 known FA DNA-repair genes. FA individuals have an increased risk of head-and-neck squamous-cell-carcinomas (HNSCC), often fatal. Systemic intolerance to standard cisplatin-based protocols due to somatic-cell hypersensitivity underscores the urgent need to develop novel therapies. Here, we performed unbiased siRNA screens to unveil genetic interactions synthetic-lethal with FA-pathway deficiency in FA-patient HNSCC cell lines. We identified based on differential-lethality scores between FA-deficient and FA-proficient cells, next to common-essential genes such as PSMC1, PSMB2, and LAMTOR2, the otherwise non-essential RBBP9 gene. Accordingly, low dose of the FDA-approved RBBP9-targeting drug Emetine kills FA-HNSCC. Importantly both RBBP9-silencing as well as Emetine spared non-tumour FA cells. This study provides a minable genome-wide analyses of vulnerabilities to address treatment challenges in FA-HNSCC. Our investigation divulges a DNA-cross-link-repair independent lead, RBBP9, for targeted treatment of FA-HNSCCs without systemic toxicity.

[1] Oncogenetics, Dept. of Human Genetics, Cancer Center Amsterdam, Amsterdam University Medical Center, Amsterdam, Netherlands. [2] OncoProteomics Laboratory, Cancer Center Amsterdam, Amsterdam University Medical Center, Amsterdam, Netherlands. [3] Dept. Epidemiology and Biostatistics, Amsterdam University Medical Center, Amsterdam, Netherlands. [4] Dept. of Medical Oncology, Cancer Center Amsterdam, Amsterdam University Medical Center, Amsterdam, Netherlands. [5] Dept. of Otolaryngology, Head and Neck Surgery, Cancer Center Amsterdam, Amsterdam University Medical Center, Amsterdam, Netherlands. [6] Division of Tumor Biology and Immunology, Netherlands Cancer Institute, Amsterdam, Netherlands. [7] Present address: Biostatistics facility, Netherlands Cancer Institute, Amsterdam, Netherlands. [✉]email: m.pai@amsterdamumc.nl; jc.dorsman@amsterdamumc.nl

Genome maintenance in eukaryotes requires dedicated, multi-factorial pathways to resolve a vast variety of DNA lesions[1]. The fidelity of DNA repair is known to decline with advancing age and is inversely correlated with the incidence of cancer[2,3]. In fact, mutational loss of DNA repair pathways often causes impaired cellular fitness of normal cells. This may pose a strong selection pressure for the acquisition of adaptations which favour tumour growth. Therefore, fast-growing tumour cells, in particular, are thought to be more dependent on such molecular processes than their normal non-transformed counterparts.

A crucial DNA repair network is the Fanconi anaemia (FA) pathway. FA proteins resolve defined lesions posing as impediments to the replication fork, termed DNA inter-strand crosslinks (ICLs)[4]. Caused by the mutational loss of one of the 22 currently-identified FA genes[5], FA is a rare recessive chromosomal-instability syndrome, resulting in hypersensitivity to DNA cross-linkers and proliferative impairment, particularly in stem cells and otherwise rapidly dividing cells[6,7]. With an incidence of 1:200,000, FA individuals clinically present with stunted growth and congenital anomalies, which reflect the affliction of normal cell division during early development. FA patients are predisposed to blood malignancies (AML) and squamous cell carcinomas of head and neck (HNSCC) and anogenital region, often very aggressive[8,9]. The intolerance of the FA-defective somatic cells to cisplatin and radiation limits treatment options, especially in advanced FA-HNSCC; the major cause of death in adult FA patients now. This imposes an urgent need to develop novel therapeutic strategies. Understanding how FA-deficient cancer cells acquire increased cellular fitness in the course of oncogenic transformation could be a foothold to develop tailored therapeutics. Unbiased functional genomic CRISPR/Cas9 and siRNA screens are used as discovery tools[10,11] to dissect not only fundamental biological processes and novel regulators contributing to them, but also to uncover the cellular essentialomes[12–14] or context-specific synthetic lethal genetic interactions[15–19]. Here, we aimed to identify genetic interactions synthetically lethal with FA-pathway deficiency in the context of HNSCC cells.

Previously described FA-patient-derived HNSCC cell lines from two FA-complementation groups, harbouring defects in proteins functioning within the upstream FA core complex, namely FA-C and FA-L, along with one FA-complemented cell line[20–22] were used in multiple, whole-genome siRNA library screens. Based on both computed lethality coefficients and differential-lethality analysis comparing FA-deficient vs FA-complemented cells, we identified LAMTOR2, PSMB2, PSMC1 and RBBP9 as hits. Here we highlight RBBP9, a metabolic serine hydrolase[23] also shown to be crucial for pancreatic cancer cell survival, yet non-essential in several cell types and mice, as an attractive druggable candidate for FA-HNSCC treatment.

## Results and discussion

**siRNA genome-wide screens and hit calling.** To identify dependencies of FA-defective head-and-neck squamous-cell carcinomas (FA-HNSCCs), previously described[20–22] FA-patient-derived HNSCC cell lines from two FA-complementation groups VU-SCC 1131 (FA-C) and VU-SCC 1604 (FA-L), and the FANCC-complemented cell line VU-SCC 1131 (FA-C) + FANCC were used in multiple whole human genome siRNA library screens (Fig. 1 and Supplementary Table 1). Besides cells from complementation groups C and L, cells from the FANCA complementation group were used for further analyses, too; thereby focusing on the FA core complex.

Each arrayed screen interrogated currently annotated genes using 21,121 SMARTpools of 4 distinct siRNAs per gene

(Dharmacon). siRNAs arrayed in 384-well plates were reverse transfected into cells that were cultured for 96 h and then assayed for viability (Fig. 1a).

Firstly, we generated a list of candidate genes whose knockdown was lethal to the FA-defective HNSCC cell lines VU-SCC-1131 (FA-C) and VU-SCC-1604 (FA-L), indicative of common dependencies in FA-HNSCC. Two-parametric cut-offs (e.g. a model coefficient $\geq 0.8$ means strongly lethal and FDR $\leq 10^{-8}$ indicates highly statistically significant) were used to generate the lists of lethal siRNAs for each cell line. A genetic-interaction network and pathway analysis, depicting the common denominators of lethality in FA-HNSCC, is presented in Fig. 1b (see also Supplementary Fig. 1a, b and normalised screen data in Supplementary Data 1). These genes mainly belong to a group of already known common-essential genes involved in protein homoeostasis. Contrary to the well-studied example of *BRCA*-mutated tumours being sensitive to PARP inhibition[24–26], our screens did not reveal any synthetic-lethal relationship with PARP/PARG family (Fig. 1c). Other FA/BRCA genes did not display synthetic lethality. This analysis yielded, nevertheless, yet unidentified candidate genes such as RBBP9, thus highlighting the power of genuinely unbiased target discovery approaches.

In pursuit of hits that are specific to the FA-pathway-deficient tumour context, we further refined our analysis by stringently filtering for candidate genes whose knockdown was highly lethal to FA-C VU-SCC-1131 cells but not to the corrected cell line VU-SCC 1131 + FANCC. The resulting list was cross-referenced to check for lethality in the FA-L VU-SCC 1604 cell line (Venn diagram in Fig. 2a). From this analysis, eight genes, viz. RBBP9, LAMTOR2, EIF4A3 (DDX48), RPL29, PSMB2, PSMC1, FLJ13150 (RPAP2) and C21ORF49 were found to be essential in the two FA-deficient HNSCC cell lines but not when FANCC was restored, i.e. their knockdown manifested lethality in a sought-after manner (Fig. 2b). The putative unknown protein C21ORF49 was not further pursued, because this gene is now considered to be an RNA gene, and is affiliated with the lncRNA class. Knockdown of five other genes, namely C13ORF3, CRSP2, KIF23, LOC401466 (C8ORF59) and NUP153 was lethal to FA-proficient, but not FA-deficient HNSCC tumour cells (Supplementary Fig. 2a, heatmap of top 50 differentially lethal genes in the FA-C cell line pair compared to the FA-L and sporadic VU-SCC-120 cell line).

**RBBP9 is synthetic lethal in FA-deficient HNSCC.** We next validated the eight potential FA-HNSCC-specific candidate hits in independent cell viability experiments under experimental settings similar to the primary screens. Four genes viz., RBBP9, LAMTOR2, PSMB2 and PSMC1 could be validated in cell viability assays (Fig. 2c), while three others, namely EIF4A3 (DDX48), RPL29 and RPAP2 were inconsistent with the primary-screen data and hence not further considered (Supplementary Fig. 2b). As an essential control, the four siRNAs targeting the genes of interest were tested by scoring their efficacy. When at least two of four single siRNAs targeting the gene of interest reproduced the lethal effect of their cognate siRNA pools in the screened cell lines, the candidate gene was considered a genuine hit (Supplementary Fig. 2c). Specific silencing of all four validated target genes was confirmed at the transcript level by quantitative PCR (Supplementary Fig. 3a) and at the protein level by immunoblotting (Supplementary Fig. 3b and uncropped blots in Supplementary Fig. 7) in all four FA-HNSCC cell lines within 24 h of knockdown with a final concentration of 10 nM siRNA.

Among the validated target genes, retinoblastoma binding protein 9 (RBBP9) attracted our attention for several reasons. Firstly, this poorly characterised small protein was thought to be

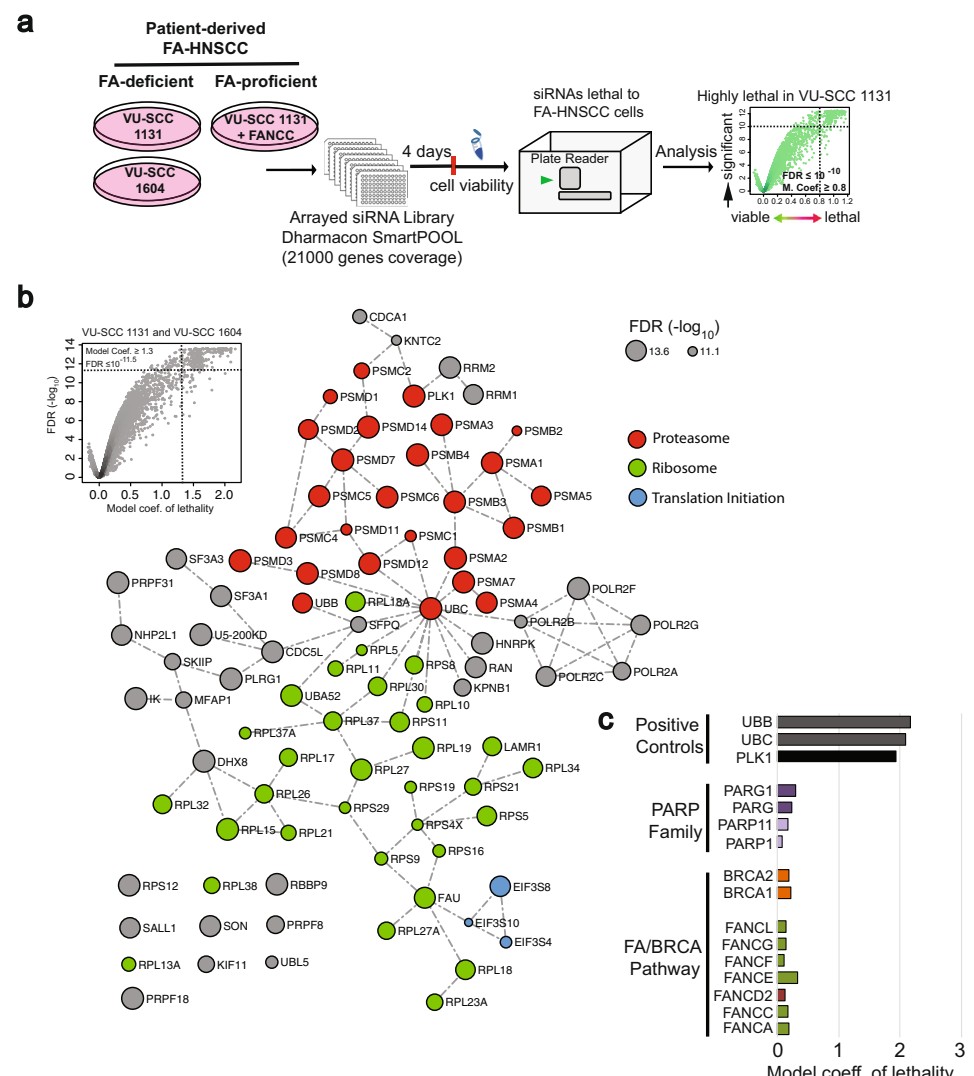

**Fig. 1 Genome-wide synthetic lethality screens in Fanconi anaemia head and neck squamous-cell carcinoma. a** Design and format of genome-wide siRNA screens performed in patient-derived FA-HNSCC cell lines VU-SCC 1131 (VU1131) from complementation groups FA-C and VU-SCC 1604 (VU1604) from complementation group FA-L along with a phenotype-corrected (wild-type FANCC complemented) cell line VU-SCC 1131 + FANCC (VU1131 + FANCC) as a reference to delineate genetic-interaction networks synthetic lethal with the loss of FA pathway. Screens were performed by reverse transfecting cells in 384-well plates previously arrayed with siRNA library (25 nM each, Dharmacon siARRAY Whole Human Genome) and cell viability was assessed using the CellTiter-Blue® assay 4 days after gene silencing. Fluorescence signals measured at 570 nm were log-2 transformed and normalised using the RSCREENORM approach (see ref. 24, main text) and lethality scores were calculated per gene, for analysis. All screens were performed in triplicates. Volcano plots depict model coefficient of lethality (X-Axis) = statistically derived relative lethality score, values approaching or above 1.0 = highly lethal, and false discovery rate FDR (Y-Axis) = statistical significance. All screens were performed in triplicate. **b** Gene essentiality, interaction network and pathway analysis of combined lethal vulnerabilities identified in relation to the non-targeting siRNA and positive controls, in FA-C and FA-L HNSCC cell lines VU-SCC 1131 and VU-SCC 1604. Genetic interactions are mined from the GeneMania database and pathway predictions are generated using Reactome. Nodes and edges represent gene names and known physical interactions, respectively. Node sizes are scaled to −log10 FDR values. Node colours depict pathway associations. Note that the red nodes imply the involvement of the protein degradation machinery at large, and are not necessarily limited to APC/C-mediated protein degradation in mitosis. Also, see Supplementary Fig. 1 for lethal interaction networks in individual FA-HNSCC complementation group cell lines. **c** Performance of a selection of genes of the FA-BRCA pathway (FA genes—A, C, D2, E, F, G and L, BRCA1, BRCA2) along with positive controls PLK1, UBB and UBC, as an indication of biological effect size upon silencing. Genes are ranked based on the model coefficient of lethality upon silencing as well as FDR. Hits with a calculated model lethality coefficient ≥1.0 were considered to have a strong lethal effect. For comparison, straight-lethal positive control siRNAs PLK1, UBB and UBC are plotted alongside (lethality coefficient ca. 2). Bars represent an average of 9 values per gene. Note that, as expected, silencing of additional FA or BRCA genes did not result in significant lethality in these FA-HNSCC lines, while the PARP family genes—well-documented to be synthetic lethal with BRCA loss—did not result in lethality either.

binding to RB1[27], and has been reported to be essential for the survival of pancreatic neoplasms in a seminal study of a large patient cohort by Shields et. al., This effect was attributed to the antagonistic effects of RBBP9 on growth inhibitory TGF beta signalling[28]. Of note, about 7–10 % of 'sporadic' pancreatic cancers harbour mutations in the FA/BRCA pathway, particularly the young-onset cases[29]. RBBP9 stands apart from the 9-member RB-binding protein family (RBBP1 to RBBP9/RBBP10) in its function as a carboxylic ester hydrolase. Despite these findings, RBBP9 is still categorised as an orphan enzyme.

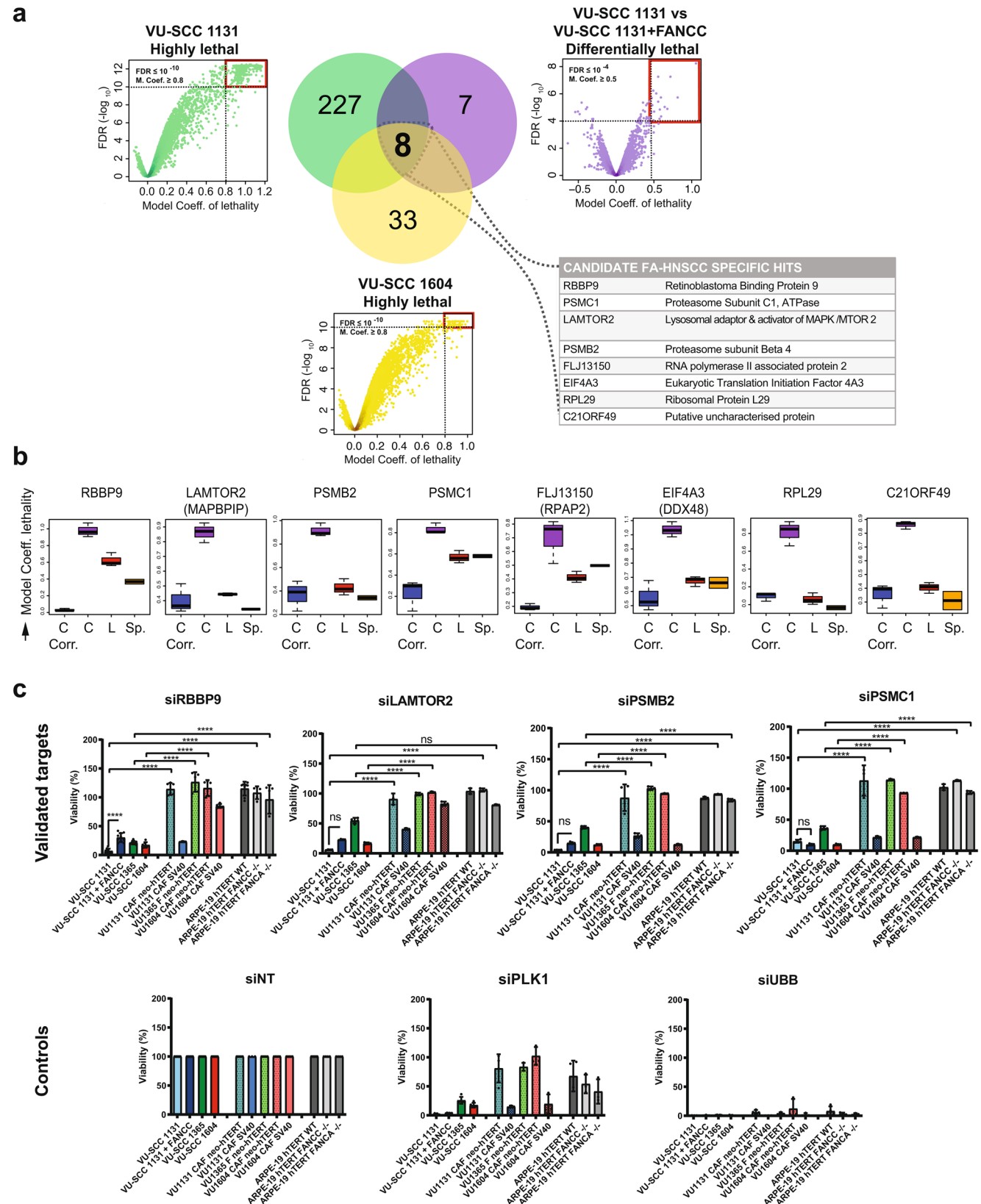

**a**

VU-SCC 1131 Highly lethal

VU-SCC 1131 vs VU-SCC 1131+FANCC Differentially lethal

VU-SCC 1604 Highly lethal

| CANDIDATE FA-HNSCC SPECIFIC HITS | |
|---|---|
| RBBP9 | Retinoblastoma Binding Protein 9 |
| PSMC1 | Proteasome Subunit C1, ATPase |
| LAMTOR2 | Lysosomal adaptor & activator of MAPK /MTOR 2 |
| PSMB2 | Proteasome subunit Beta 4 |
| FLJ13150 | RNA polymerase II associated protein 2 |
| EIF4A3 | Eukaryotic Translation Initiation Factor 4A3 |
| RPL29 | Ribosomal Protein L29 |
| C21ORF49 | Putative uncharacterised protein |

**b**

**c**

Secondly, RBBP9 is small, ubiquitously expressed, highly conserved[30] and largely non-essential in a wide variety of cell types (as judged by data from minable public CRISPR databases; https://orcs.thebiogrid.org/ and https://depmap.org/portal/). We reasoned that RBBP9 may offer a broad therapeutic window, as opposed to LAMTOR2, PSMB2 and PSMC1, which are essential

genes indispensable to cell survival and growth upon knockout (See also BioGRID ORCS and https://depmap.org/portal/).

**RBBP9 silencing is toxic to transformed, and not to untransformed FA-deficient cells.** Thus far, our analysis led to the

**Fig. 2 Identification of FA-HNSCC specific target genes, target validation and hit selection. a** Analysis strategy to enrich for FA-HNSCC specific hits—Shows stringent filtering for candidate genes whose knockdown was both highly lethal to FA-C VU-SCC-1131 cells with cut-offs, model coefficient of lethality ≥0.8, FDR ≤$10^{-10}$ as well as differentially lethal to the FA-C cell line pair, i.e. VU-SCC 1131 and VU-SCC 1131 + FANCC using a model coefficient of lethality ≥0.5, FDR ≤$10^{-4}$. Being also represented among genes highly lethal in the FA-L VU-SCC 1604 tumour cell line, model coefficient of lethality ≥0.8, FDR ≤$10^{-10}$, a subset of eight genes shown in **a**, table inset, emerged as FA-HNSCC specific. model coefficient = mathematically derived relative lethality score, values approaching or above 1.0 = highly lethal, FDR false discovery rate, indicates statistical significance. **b** Box plots of data from three replicates of the primary screen, indicating calculated, relative lethality scores (model coefficient, Y-Axis) of candidates identified as described in **a**. C = VU-SCC 1131, C corr. = VU-SCC 1131 + FANCC, L = VU-SCC 1604. Bars corresponding to Sp. sporadic HNSCC cell line VU-SCC 120 screened previously[61]. See also Supplementary Fig. 2. Mid-line = median, top and bottom hinges correspond to the third and first quartiles, respectively. Error bars are ±1.58 IQR/sqrt(n), where n number of samples and IQR inter-quartile range i.e. third – first quartile. **c** Validated targets (**c**, upper row) discovered in the primary siRNA screen of FA-HNSCC tumour cell lines, along with controls (**c**, lower row), in an extended panel of patient-matched fibroblasts (Fs) or cancer-associated fibroblasts (CAFs) as indicated, as well as human diploid retinal pigment epithelial ARPE-19 hTERT cell lines with a targeted disruption of the FANCA or FANCC genes, all representing FA-defective untransformed cells. siRNA deconvolution, qRT-PCR and immunoblot validation of On-Target knockdown efficiency is presented in Supplementary Figs. 2, 3. Note that hTERT and SV40-immortalised patient Fs/CAFs respond differently to the target gene knockdowns. Also, note that UBB is a robust 'killer' control compared to PLK1 silencing. All siRNAs were used at a concentration of 10 nM in the validation experiments. CellTiter-Blue® was used as a readout for viability. Fluorescence values were normalised to untreated controls of the respective cell line. RBBP9 is the only non-essential gene of interest as an FA-HNSCC-specific hit. n = 3 independent experiments. Error bars = SD. controls - NT non-targeting, UBB ubiquitin B, PLK1 Polo-like kinase 1. Non-parametric two-way ANOVA with Holm–Sidak correction for multiple comparison was performed for statistical analysis. **$p ≤ 0.01$, ****$p ≤ 0.0001$. See also Supplementary Fig. 3.

---

discovery of hits that were lethal to FA-defective HNSCC cell lines. Unbiased efforts to discover genes essential in FA-mutated tumours is indeed valuable, but only as a first step. The true concern lies in the constitutionally FA-gene-mutated somatic cells that make patients intrinsically vulnerable to many interventions as a consequence of the defect.

In order to accommodate and account for the somatic-cell hypersensitivity of FA patients, we included FA-HNSCC patient-matched fibroblasts from the skin (Fs) or tumour stroma (CAFs, cancer-associated fibroblasts), as well as FA-mutant ARPE-19 cells in the validation experiments (Fig. 2c). We observed that RBBP9 knockdown resulted in profound lethality in FA-defective tumour cells, while lethality in FA-proficient HNSCC cells was less severe i.e. differentially lethal. Importantly, RBBP9 silencing was significantly less toxic to FA-defective, patient-matched CAFs or Fs, as well as to CRISPR-Cas9 engineered human diploid WT, $FANCC^{-/-}$ and $FANCA^{-/-}$ ARPE-19 hTERT cell lines. In fact, this trend holds true for all four targets tested, namely RBBP9, LAMTOR2, PSMB2 and PSMC1. Interestingly, the level of toxicity of target gene silencing in patient-derived CAFs/Fs, was higher in cells transformed by Simian Virus 40 (SV40) than in cells immortalised by human telomere-reverse-transcriptase (hTERT). This indicates that lethality caused by genetic inhibition of RBBP9 was stronger in transformed FA-defective cells than in FA-defective untransformed cells, suggesting that RBBP9 targeting may be valuable for FA-HNSCC treatment.

At least one study reports an *Rbbp9* knockout mouse model with no obvious developmental phenotype[31], while our own work demonstrates that gene inactivation in normal diploid human cells appears to cause no apparent harm to the cell survival and growth (See also BioGRID ORCS and depmap portal). Although not abundant, RBBP9 expression appears to be consistent among metabolically active and secretory tissue types such as germ cells, stem cells, pancreatic cells and oral squamous cells (https://www.proteinatlas.org/ENSG00000089050-RBBP9/pathology).

**Knockdown of RBBP9 is detrimental to FA-HNSCC cell proliferation and induces apoptosis.** To study the cell biological consequences of RBBP9 silencing (structural features in Fig. 3a), we performed proliferation analyses of the FA-HNSCC cell lines, along with the untransformed cell line panel, in the IncuCyte™ S3. We observed a general, severe debility in proliferation in all FA-HNSCC cell lines when RBBP9 was knocked down compared to the non-targeting control, whereas the FA-proficient VU-SCC

1131 + FANCC continued to thrive under these conditions (Fig. 3b, c), suggesting that RBBP9 function is required for FA-HNSCC cell proliferation. Interestingly, attempts to generate stable cell lines expressing RBBP9-OFP or RBBP9-FLAG were not successful in FA-HNSCC lines, suggesting that chronic abundance of RBBP9 may also have deleterious consequences in FA-HNSCC cell lines. In order to establish the cause of failure to thrive under these conditions, we followed FA-HNSCC cell lines by live-cell imaging with the NucView® 488 Caspase-3 fluorogenic substrate assay after RBBP9 silencing for 100 h. We noted not only Caspase-3 dependent cell death, but also distinct kinetics of Caspase-3 activity in the different FA-complementation group tumour cell lines (Fig. 3d, e and see profiles of Caspase-3 substrate fluorescence, also see Supplementary Fig. 4). Importantly, cells from all three complementation groups display apoptotic type of death, although with different kinetics. Moreover, the loss of plasma membrane integrity and terminal loss of cell viability was confirmed by Propidium Iodide (PI) exclusion flow cytometry (Fig. 3f, g). These data thus suggest canonical Caspase-3-mediated apoptosis induction as the major mode of cell death when RBBP9 was downregulated.

**Pharmacological inhibition of RBBP9 partially phenocopies genetic silencing.** We next sought to know whether the RBBP9 inhibitor Emetine hydrochloride (Fig. 4a) may have therapeutic value in FA-HNSCC. Emetine has been reported to be a selective RBBP9 inhibitor[32], although with a plethora of biological effects, including anti-viral[33] and antitumour activities in a variety of cancer types[34–40]. We observed that at a low treatment dose of 15 or 25 nM, Emetine selectively killed HNSCC tumour cell lines, but had no apparent effects on patient-matched CAFs and Fs as well as untransformed ARPE-19 hTERT cell lines with targeted disruption of the FANCA and FANCC genes (Fig. 4b, c). Consistently, genetic disruption of RBBP9 in untransformed cells (Supplementary Fig 5a–c, uncropped blots pertaining to Supplementary Fig. 5c in Supplementary Fig. 8) did not affect cell viability (Fig. 4b–d), cell cycle (Supplementary Fig. 5d and Gating strategy in Supplementary Fig. 9) or growth kinetics (Supplementary Fig. 5e). We further confirmed the potency of Emetine in 3D cultures in the HNSCC tumour cell lines, as well as in the FA-defective untransformed cells. We found a significant volume reduction of Emetine-treated FA-HNSCC spheroids compared to their untreated counterparts, while untransformed cells appeared to be unaffected by Emetine (Fig. 4d). Emetine-induced tumour

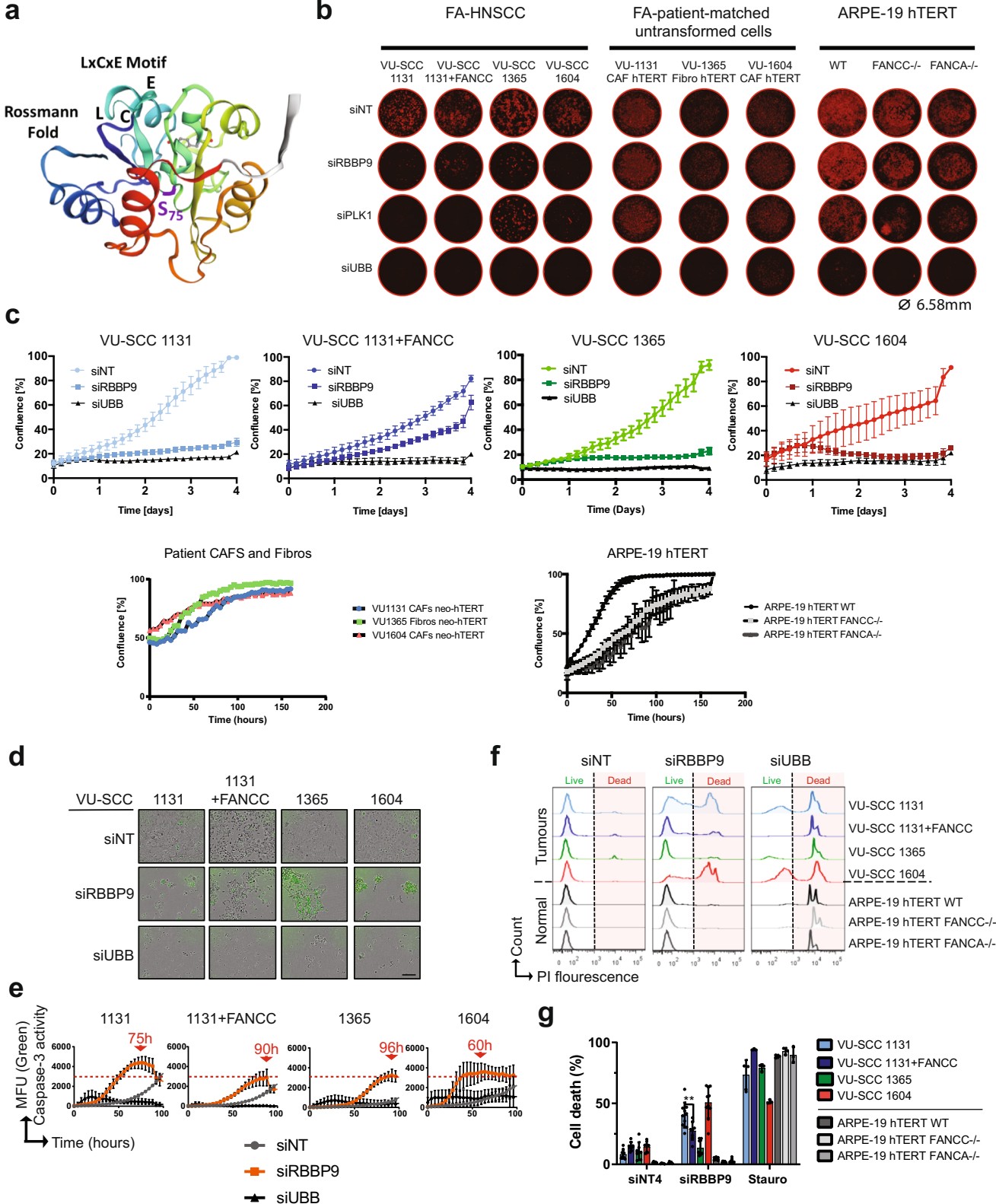

spheroid volume reduction seems to be at least in part due to apoptosis, as observed from NucView 488 Caspase-3 fluorogenic substrate assays within 24 h after emetine treatment (Supplementary Fig. 5f).

Although previously shown to be selective for RBBP9 within the family of metabolic serine hydrolases, Emetine has also been shown to affect other relevant cellular processes, viz. lagging

strand DNA synthesis inhibition[41] by mechanisms not completely understood, albeit at significantly higher, acute doses than those used in our experiments. Protein translation inhibition has also been reported, which can already occur at lower doses[42]. Immunoblot analysis of 25 nM Emetine-treated VU-SCC-1131 and its FA-corrected counterpart VU-SCC-1131 + FANCC showed no measurable difference in phosphorylation and

**Fig. 3 Deleterious effects of RBBP9 knockdown in FA-HNSCC. a** Ribbon representation of RBBP9 structure adapted from SWISS-MODEL (#O75884, O75884 | SWISS-MODEL Repository (expasy.org)), highlighting the reactive Serine residue (S75, in purple) of the enzyme's catalytic triad, putative RB-interacting LxCxE motif (Residues 63, 65, 67 in aqua) and the mononucleotide-binding Rossmann fold in navy blue. **b** Representative fluorescent photomicrographs of fixed, propidium iodide stained cells in individual wells of 96-well plates (4x objective, IncuCyte®) after 96 h of RBBP9 knockdown, (as in validation experiments) demonstrating its selectively lethal effects in FA-HNSCC tumour cells. Ø = 6.58 mm, well diameter. **c** Confluence-based growth kinetics of FA-HNSCC tumour cell lines with RBBP9 knockdown for 96 h (top row). Note the higher proliferation rate of FA-reconstituted (hence ICL-repair proficient) VU-SCC 1131 + FANCC cell line, indicating conditional synthetic lethality of RBBP9 with FA-pathway loss. Similar growth curves of respective FA patient-matched hTERT-immortalised CAFs/Fs and the ARPE-19 hTERT WT, FA-A and FA-C model cell lines are shown in the lower row for comparison. Despite similar seeding densities, note the lag in growth curves of both FANCC−/− and FANCA−/− ARPE-19 hTERT cell lines, typical of cells harbouring FA-defects. n = 3 independent experiments in duplicates. Error bars = SD. NT non-targeting, UBB ubiquitin B, PLK1 Polo-like kinase 1. **d, e** Representative photomicrographs (at 72 h) of live FA-HNSCC cells IncuCyte®-imaged in 96-well plates for upto 100 h after RBBP9 knockdown, fed with the Caspase-3 fluorogenic substrate NucView 488®. Green fluorescent cells represent apoptosis. Caspase-3 activity maxima plotted as green object counts over time (in **e**, upto 100 h), reflect cell death as a function of time, indicating differences between the FA patient-derived HNSCC cell lines. n = 3 independent experiments in duplicates. Error bars = SD. Scale bar ≈150 µm. See also Supplementary Fig. 4 for full-frame images with scale bars. **f, g** Propidium Iodide exclusion flow cytometry plots of the FA-HNSCC and FA-normal ARPE-19 hTERT lines (quantification in **g**) to measure terminal cell death. Bars are a percentage of PI-positive (Dead) cells. RBBP9 knockdown (72 h in these experiments) does not affect untransformed ARPE-19 hTERT cells with or without FA-defects. n = 3 independent experiments in triplicates. Error bars = SD. Non-parametric two-way ANOVA with Holm–Sidak correction for multiple comparison was performed for statistical analysis. **p ≤ 0.01.

---

activation of the replication checkpoint kinases ATR and Chk1, but also modestly reduced phosphorylation of the DNA damage responsive phospho-Ser 139 H2AX (Fig. 4e, uncropped blots in Supplementary Fig. 6), suggesting that Emetine-induced cell death is independent of DNA replication impediments. Together, these data provide first-hand evidence, that low-dose Emetine has strong antitumour effects while causing no harm to normal cells with FA-defects.

**Relation of RBBP9 to other hits**. In spite of RBBP9 being reported in several biochemical serine hydrolase activity studies[30,43,44], RBBP9 remains an elusive, poorly characterised cellular protein with no identified biological substrate(s). To date empirical protein–protein interaction studies are limited. In order to gain an understanding of its biological function(s), we resorted to RBBP9-FLAG over-expression followed by co-immunoprecipitation-mass spectrometry in HEK293T cells, with FLAG-FANCC being used as a control. Among the 32 high-confidence RBBP9-interacting proteins short-listed on the basis of spectral count abundance and p value (≤0.05), we found proteins vital to DNA replication and genome maintenance, such as RECQL5, MCM5 and RIF1. Four interacting proteins, namely SNRPD3, PRPF31, RPS11 and RPS15, all involved in pre-mRNA processing, splicing and protein synthesis, were found to be essential in the original FA-HNSCC screens and are known essential genes (network in Fig. 4f and Mass Spec data in Supplementary Data 2).

It is noteworthy that validated candidate hits from this study, namely LAMTOR2, PSMB2 and PSMC1 are principally genes functioning directly or indirectly in the protein homoeostasis network. Interestingly, RBBP9 has also recently been linked to proteasome function[45]. Combined, these data support the idea that relieving proteotoxic stress may be especially important in HNSCC tumours in FA individuals.

**Screen data in relation to genetic studies on HNSCC and CRISPR screens**. Recent works[46–49] based on mapping the genomic landscape of sporadic head-and-neck cancers have implicated PI3K, NOTCH and HIPPO pathways as oncogenic drivers and furthered our understanding of the disease. A study by ref. [50], using genome-wide CRISPR screens shows that DNA polymerase iota compensates for DNA replication-related deficiencies due to loss of FA pathway in a colon carcinoma model with a targeted disruption of FA genes. However, these genes did not show up as major hits on our screen (Supplementary Data 3).

**Conclusions**. Our work provides a publicly available dataset to understand the in vitro liabilities in the context of a rare, inherited genomic instability and cancer-causing disease. Especially the use of siRNA screens, compared to KO CRISPR screens, may represent a picture closer to clinical translation and practice [10, 52]. Moreover, the limited sample size due to the rarity of the syndrome makes this study both challenging as well as unique.

Next to common-essential genes such as PSMC1, PSMB2 and LAMTOR2, all linked to protein homoeostasis, the otherwise non-essential RBBP9 serine hydrolase stands out as the prime targetable hit in FA-HNSCCs. Based on our own data and published reports on RBBP9, combined with the knowledge on the mode of action of low-dose Emetine, also a relation with protein homoeostasis may well be forthcoming. Our data furthermore underscore that unbiased discovery tools may reveal targets thus far unanticipated leads to combat tumours that are difficult to treat. In vivo experiments employing e.g. xenograft tumour models in mice will be required to further confirm the translational potential of RBBP9 inhibition for FA-HNSCC treatment. In conclusion, our study identifies a DNA cross-link repair independent lead, RBBP9, for targeted intervention in FA-HNSCCs that may spare normal somatic FA cells.

## Methods

**Cell lines and cell culture**. Previously described[22] FA patient-derived HNSCC cell lines VU-SCC 1131 (FA-C) and VU-SCC 1604 (FA-L), along with a lentiviral FANCC-complemented cell line (VU-SCC-1131 + FANCC) were used in genome-wide screens. The generation of the cell lines for scientific purposes was approved by the Institutional Review Board at the time[22]. FA-HNSCC patient-derived tumour cell lines were maintained on DMEM (GIBCO, Cat# 11330057) supplemented with 10% Foetal Bovine Serum (GIBCO, Cat#10270). VU-SCC-1131 + FANCC was maintained on 1ug mL⁻¹ puromycin (SIGMA, Cat#P7255; selection for FANCC expression). Primary human retinal pigment epithelial cell line ARPE-19 (ATCC, Cat# CRL-2302) was immortalised by electroporating with pBABE-hygro-hTERT construct (Addgene, Cat# 1773) followed by hygromycin-B selection. ARPE-19 hTERT cell lines were maintained on DMEM-F12 (GIBCO, Cat# 31331028) supplemented with 10% FBS with 15 ug ml⁻¹ Hygromycin-B (Roche, Cat#10843555001). hTERT-immortalised FA-Patient-matched Cancer-Associated Fibroblasts (CAFs) and Skin Fibroblasts (Fs) were maintained on IMDM (GIBCO, Cat# 31980030) supplemented with 10% FBS and 2x Embryomax Nucleoside supplement (Merck, Cat# ES-008-D). SV40-immortalised FA-patient-matched cancer-associated fibroblasts (CAFs) and skin fibroblasts (Fs) and HEK293T cells were maintained on DMEM + 10% FBS. For 3D cultures, single spheroids consisting of 5 × 10³ FA-HNSCC tumour cell lines and ARPE-19 hTERT cell lines were grown in DMEM-F12 without FCS, supplemented with 5 ng ml⁻¹ Epidermal growth factor—EGF (GIBCO, Cat# PHG0311L), 5 ng ml⁻¹ basic fibroblast growth factor—bFGF (Reliatech, Cat# RLT-300-002) and 1x B27 growth supplement (GIBCO, Cat# 17504044) in 96-well cell-repellent surface plates (Greiner, Cat# 655970). All cells were maintained at 37 °C, 5% CO₂. Cell lines were routinely outsourced to Microbiome T.a.v, Amsterdam, for PCR-based

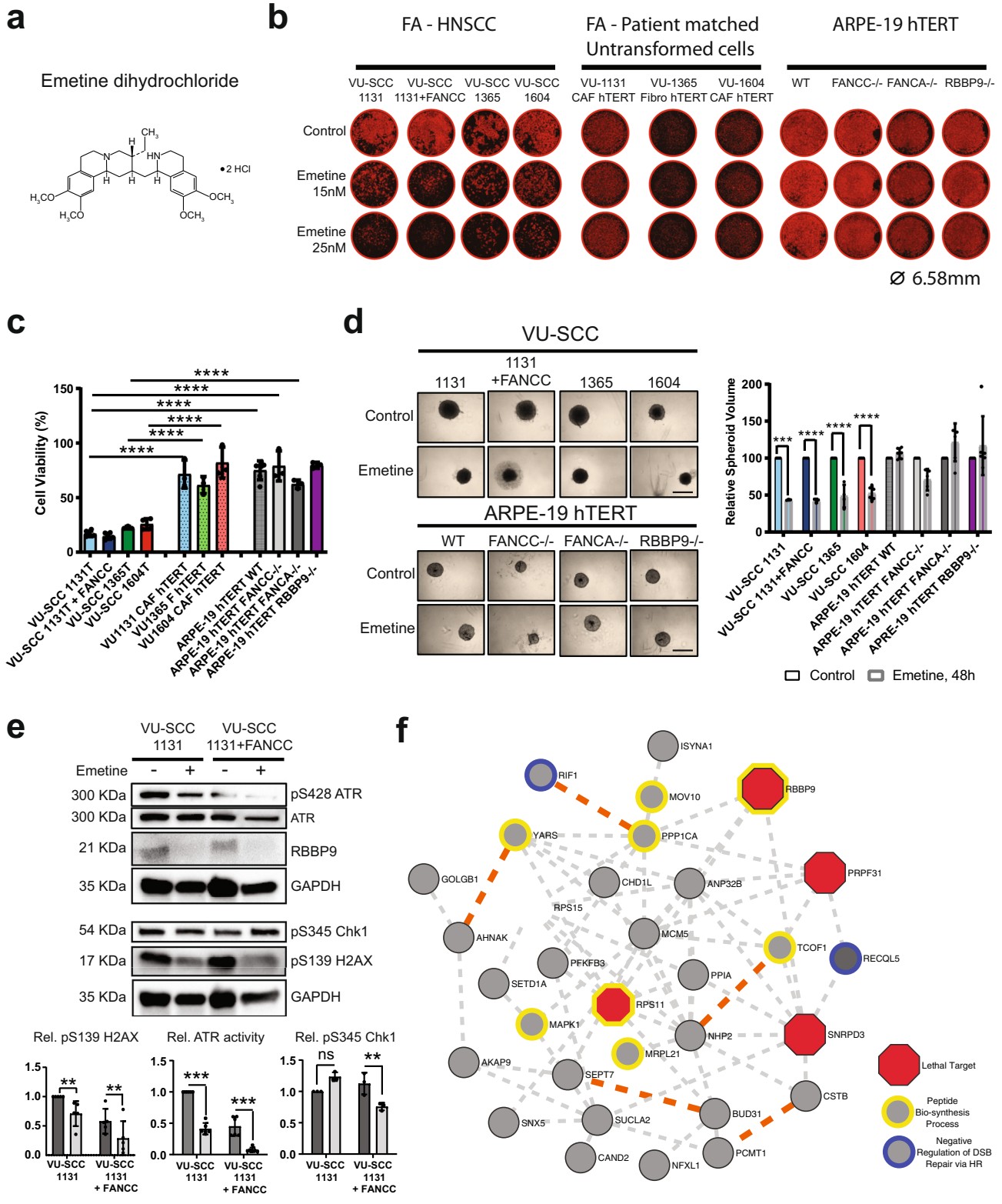

mycoplasma testing and confirmed to be negative. See Supplementary Table 1 for a list of cell lines used in this study.

**Generation of CRISPR knockout cell lines**. RBBP9 knockout ARPE-19 hTERT cells were generated by electroporation of a Cas9:tracrRNA:crRNA ribo-nucleoprotein (RNP) complex sequentially. Briefly, the RBBP9 crRNA GTGAC-CACCCACGGCTGGTA (200 µM) and Alt-R CRISPR-Cas9 tracrRNA (200 µM) (IDT, Cat# 1075928) oligos were mixed in equimolar concentrations to a final

duplex concentration of 44 µM in a 10 µL volume of IDTE Buffer (IDT, Cat# 11-01-02-02), annealed by heating to 95 °C for 5 min and cooling to RT. For each electroporation, 0.3 µl of Alt-R Cas9 nuclease (61 µM) (IDT, Cat# 1074182) was combined with 3 µl of crRNA:tracrRNA duplex in 7.7 µl of IDT buffer R and gently mixing to achieve a final Cas9:guideRNA ratio of 1:4. The mix was incubated at RT for no longer than 10 min. 4 µl of the RNP-Cas9 complex was mixed with 75,000 cells in 7 ul volume of Buffer R and 1 µM Alt-R Cas9 electroporation enhancer (IDT, Cat# 1075916) in a final volume not exceeding 15 µL. This mix was pulsed in a pre-set Neon™ nucleofection device (Thermo Fischer, Cat# MPK5000) at 1400 V

**Fig. 4 Pharmacological targeting of RBBP9 enzymatic function in FA-HNSCC. a** Chemical structure of the known irreversible RBBP9 inhibitor Emetine, an oxime ester, drawn using adobe illustrator. **b** Representative photomicrographs (right) of propidium iodide stained cells in individual wells of 96-well plates (4x objective, IncuCyte®) after 96 h of Emetine (15 and 25 nM) treatment, demonstrating its selectively lethal effects in FA-HNSCC tumour cells. ARPE-19 hTERT WT, FANCC$^{-/-}$ and FANCA$^{-/-}$ cells were included to denote human, diploid, untransformed cells with or without FA-proficiency, respectively, while ARPE-19 hTERT RBBP9$^{-/-}$ cells is a control to monitor any gross off-target lethality due to Emetine. Ø = 6.58 mm, well diameter. Generation and characterisation of ARPE-19 hTERT RBBP9$^{-/-}$ cells is presented in Supplementary Fig. 5. **c** Cell viability (CellTiter-Blue®) assay of the cell line panel as in **a**, after 96 h of 25 nM Emetine treatment. Note the general lethality in HNSCC cells, while all untransformed cells, patient-matched or otherwise, irrespective of their genotype, remain unaffected. Fluorescence values were normalised to untreated controls of the respective cell line. $n = 3$ independent experiments in duplicates. Error bars = SD. Non-parametric two-way ANOVA with Holm–Sidak correction for multiple comparison was performed for statistical analysis. **$p ≤ 0.01$, ****$p ≤ 0.0001$. **d** Representative photomicrographs (left) and the mean spheroid volumes (right) of control or Emetine-treated (500 nM, 48 h) treated 3D spheroid cultures of FA-HNSCC tumour cell lines, as well as human diploid retinal pigment epithelial ARPE-19 hTERT cell lines with a targeted disruption of the FANCA and FANCC genes, representing FA-defective untransformed cells. RBBP9$^{-/-}$ ARPE-19 hTERT cell line served as a control for Emetine's activity. $n = 2$ independent experiments in triplicates. Error bars = SD. Scale bar = 200 um. Non-parametric two-way ANOVA with Holm–Sidak correction for multiple comparison was performed for statistical analysis. ***$p ≤ 0.001$, ****$p ≤ 0.0001$. **e** Immunoblot analysis (top) and densitometric quantification (bottom) of the isogenic VU- SCC 1131 (FA-deficient) and VU-SCC 1131 + FANCC (FA-proficient) cell line pair treated with 25 nM Emetine for 72 h. Levels of the DNA damage marker pS139 H2AX, and the activity of replication checkpoint kinases pS428 ATR and pS345 Chk1, collectively suggest that low-dose Emetine treatment induces cell death independent of DNA damage. $n = 3$ independent experiments. GAPDH was used as the loading control. Error bars = SD. **$p ≤ 0.01$, ***$p ≤ 0.001$. **f** Proposed biochemical network of RBBP9 based on protein–protein interaction(s) identified by co-immunoprecipitation (coIP) and MS analysis RBBP9-FLAG overexpressed in HEK293T cells. High-confidence interactors were shortlisted based on statistics (p value ≤0.05) and further enriched based on higher peptide count representation in RBBP9-FLAG co-IPs compared to FANCC-FLAG co-IPs. Dotted orange lines denote known physical interactions, and grey dotted lines co-expression. Functional annotation was performed using the online GO analysis tool Toppgene. Benjamini FDR ≤0.05 was deemed significant.

for 20 ms, the mix was quickly transferred into 1-well of a 6-well plate with pre-warmed DMEM, and grown under standard conditions. Gene-targeting efficiency in the CRISPRed cell pool was estimated by Sanger-sequencing PCR amplicons of the target region and Synthego ICE analysis (https://ice.synthego.com/#/) after 1 week of electroporation. To derive pure cell lines, single-cell clones obtained by limiting dilution in 96-well plates were subsequently screened for out-of-frame deletions by Sanger-sequencing PCR amplicons of a target region. Loss of protein expression in candidate clones was confirmed by western blotting. FANCC$^{-/-}$ and FANCA$^{-/-}$ cells were generated in a similar manner using custom-designed crRNA oligos and will be described elsewhere. The conventional Mitomycin C sensitivity test was also used as a phenotypic marker for FA-pathway abrogation.

**Genome-wide siRNA screens.** Four FA-HNSCC cell lines VU-SCC 1365 (FA-A), VU-SCC 1131 (FA-C), VU-SCC 1604 (FA-L) and VU-SCC 1131 + FANCC were subjected to high-throughput screening using the siARRAY Whole Human Genome library (Dharmacon / Thermo Fischer Scientific, #G-003500 (Sept05), #G-003600 (Sept05), #G-004600 (Sept05) and #G-005000 (Oct05)) consisting of 21,121 siRNA pools. A single screen consisted of $68 × 384$-well plates. All screens were performed in triplicate, except VU-SCC 1365 ($n = 2$). About 25 nM Smart-POOL was arrayed onto single wells of 384-well plates (CELLSTAR, Greiner, Bio-One) using the automated robotics platform—Sciclone ALH 3000 workstation (Caliper LifeSciences) and a Twister II microplate handler (Caliper LifeSciences) and stored until use. A total of 0.01 µl per well Lipofectamine RNAimax reagent (Invitrogen, Cat# 13778150) in low-serum OptiMEM medium (GIBCO, Cat# 31985047) was dispensed using the aforementioned devices into the wells shortly before seeding the cells.). A total of $0.75–1 × 10^3$ cells per well were reverse transfected in a total volume of 60 µl in each well using the µFill microplate dispenser (Bio-Tek). The non-targeting siCONTROL#4 (herein designated siNT4) was used as a negative control, siPLK1 and siUBB SMARTpools as positive controls in twelve wells, four wells each, on every plate. Plates were incubated for 96 h at 37 °C/5% CO$_2$ and cell viability was measured using CellTiter-Blue® assay (Promega, Cat# G8081) using an Infinite F200 microplate reader (Tecan Group Ltd).

**Validation approach.** To validate candidate hits obtained from the primary screens, a broader cell line panel, including the FA-HNSCC lines VU-SCC 1131, VU-SCC 1131 + FANCC, VU-SCC 1365, VU-SCC 1604, their respective patient-matched cancer-associated fibroblasts (CAFs) or skin fibroblasts (Fs) as available, along with representative human diploid Wild-Type (WT), FANCA$^{-/-}$ and FANCC$^{-/-}$ ARPE-19 hTERT CRISPR-engineered cell lines were used. In some independent experiments, control human diploid skin fibroblasts, as well as appropriately complemented, isogenic lines derived from all three complementation groups (VU-SCC 1131 + EV and VU-SCC 1131 + FANCC, VU-SCC 1365 + EV and VU-SCC 1365 + FANCA, VU-SCC 1604 + EV and VU-SCC 1604 + FANCL) investigated were used (Supplementary Fig. 2b). EV denotes pIRES-Neo-Empty Vector. A set of 4 siRNAs against each hit identified in the discovery screens were purchased from Dharmacon Inc. Stock solutions were prepared by resuspending in RNase-free 1x siRNA Buffer (GE Healthcare, Cat# B-002000-UB-100) according to manufacturer's instructions and stored at −80 °C in aliquots. All discovery screen validation experiments and subsequent experiments

were performed in 96-well plates, under conditions similar to the screens. About 25 or 10 nM siRNA pools were reverse transfected into $3–5 × 10^3$ cells per well depending on the cell line and transfection parameters previously optimised, using 0.2 ul Lipofectamine RNAimax (Invitrogen, Cat# 13778150), in a 120 ul final volume. Plates were incubated for 96 h at 37 °C/5% CO$_2$ and cell viability was measured using CellTiter-Blue® assay (Promega, Cat# G8081). Four siRNAs targeting the selected genes were individually tested. Candidate hits were considered positive, when at least two out of four single siRNAs per the selected target gene were as lethal as the four pooled siRNAs in each cell line screened.

**Quantitative RT-PCR.** Total cellular RNA was isolated from about $1 × 10^6$ cells using the HighPure RNA isolation kit (Roche, Cat# 11828665001) and quality controlled by 260/280 nm OD ratios on Nanodrop UV-Vis Spectrophotometer (Thermo Fisher Scientific, # ND-ONE-W). Complementary DNA synthesis was performed with 1 µg of RNA template using iScript™ cDNA synthesis kit (Bio-Rad, cat# 1708891). qPCR amplification of the cDNA was conducted on the Light-Cycler® 480 (Roche) with LightCycler® 480 SYBR Green Master (Roche, cat# 04707516001) and gene-specific primer pairs for *RBBP9* (F—5′–ACATCAGACTT GGGGGATGA–3′, R—5′–GGGTCGTCAGTAGAGCCAAA–3′), *LAMTOR2* (F1— 5′–GTATGTATGCCAAGGAGACCGTG–3′, R1—5′–TAAGATGCCGCCA CTTGGGGTGA–3′, F2—5′–ATCCTCATGGACTGCATGGAGG–3′, R2—5′–TTG GCCTTGAGCATTCCAAAGC–3′), *PSMB2* (F1—5′–AGCTAACTTCACACGCC GAAAC—3′, R1—5′–TCATGCTCATCAGCCAGCCA–3′, F2—5′–CCTCGAC CGATACTACACCC–3′, R2—5′–ATGAAGCGTTTCTGGAGCTCCT–3′), *PSMC1* (F1—5′–CTGGAGGTGGCAAGAAGGATGA–3′, R1—5′–GCAGCATCT GGTCCCTTTGTTT–3′, F2—5′–CCCTGGTCACAGTGATGAAGGT–3′, R2—5′–CAGGATGGGTGAGAGGAAGCTC–3′), two independent housekeeping control genes *HPRT* (F—5′–TGACACTGGCAAAACAATGCA–3′, R—5′–GGTCCT TTTCACCAGCAAGCT–3′), and *TPT1* (F—5′–GATCGCGGACGGGTTGT–3′, R —5′–TTCAGCGGAGGCATTTCC–3′). The hot-start programme was as follows: initial denaturation at 95 °C for 5 min, followed by 45 cycles of 95 °C for 10 s, 55 °C for 10 s, 72 °C for 10 s. For each sample and gene, the cycle number at which the abundance of amplicons exceeded a predetermined threshold ($C_t$ value) was determined and the transcript amounts expressed relative to the HPRT and TPT housekeeping controls ($ΔΔC_t$ method).

**Cell lysis and western blotting.** Following indicated treatments, about 2 million cells were used for whole-cell extract preparation in ice-cold RIPA buffer (50 mM Tris-HCl pH-8.0, 300 mM NaCl, 5 mM MgCl$_2$, 1% NP-40, 0.5 % Deoxycholate and 0.1% SDS) supplemented with cOmplete™ EDTA-free protease (Roche, Cat# 5056489 001) and Phos-Stop phosphatase (Roche, Cat# 4906837001) inhibitor cocktails. Briefly, cell pellets were resuspended in RIPA buffer, vortexed thoroughly, incubated for 10 min on ice and cleared of cell debris by centrifugation at 13,000 rpm for 20 min. Viscous samples were sonicated prior to centrifugation. Protein amounts were quantified using Pierce® BCA Protein Assay Kit following the manufacturer's instructions (Thermo ScientificTM; #23225 and #23227). Proteins were denatured in 4X LDS sample buffer (NuPAGE, Cat# NP0007), by heating to 70 °C for 10 min. Based on the abundance of the specific proteins under study and resolution requirement, 25–50 µg of total protein was resolved on pre-

cast 4–15% (Bio-Rad, Cat# 4561084DC) or 8–16 % (Bio-Rad, Cat# 456-1104) mini-PROTEAN® TGX SDS-PAGE gels or PROTEAN-II Xi 10% gradient gels (Lonza, ProSieve, #50618) according to the manufacturer's instructions and transferred onto activated PVDF membranes (Millipore, Immobilon-P, #IPVH00010 and ISEQ00010) with Towbin buffer (25 mM Tris, 192 mM Glycine, 10% Methanol) in Bio-Rad TransBlot tank transfer systems. The Precision Plus Protein ™ Dual Color Standard (Bio-Rad, # 161-0374) was used as a molecular weight marker. Membranes were blocked in 5% milk or BSA in TBS-T (20 mM Tris-HCl, 150 mM NaCl, 0.1% w/v Tween20), and incubated with primary antibodies on roller banks at 4 °C overnight. The following primary antibodies were used. RBBP9 (Abcam, Cat# ab-157202, Santa Cruz, Cat# sc-101111), LAMTOR2 (Cell Signaling, Cat# 8145 S), PSMB2 (Enzo Life Sciences, Cat# BML-PW9300-0100), PSMC1 (Bethyl labs, Cat# A303-821A), P-ATR (S248) rabbit Ab (Cell Signaling, Cat# 2853 S), ATR (N-19), Goat polyclonal IgG (Santa Cruz, Cat# sc-1887), P-Chk1 (S345) (133D3) (Cell Signaling, Cat# 2348 S), Chk1 (2G1D5) mouse mAb (Cell Signaling, Cat# 2360 S), pS139 H2AX JBW-301 (Upstate/Millipore, Cat# 05-636), pS139 H2AX (Novus Biologicals, Cat# NB100-384), Histone H3 (Cell Signaling, Cat# 9715), β–Actin mAb (Santa Cruz, Cat# sc-47778), β Tubulin (D-10) (Santa Cruz, Cat# sc-5274), Vinculin (H-10) (Santa Cruz, Cat# 25336). All primary antibodies were used at 1:1000 dilution. Blots were washed 3x in 1X TBS-T, and probed with appropriate HRP-conjugated goat anti-rabbit (Dako, Cat# P044801-2), goat anti-mouse (Dako, Cat# P044701-2) or Rabbit anti-goat (Dako, Cat# P044901-2) secondary antibodies for 1 h at RT on roller banks at 1:5000 dilution. Blots were washed 3x in TBS-T, developed by chemiluminescent imaging with ECL prime western blotting detection reagent (ECL, Amersham, Cat# RPN2236) on a Bio-Rad system.

**Live-cell imaging, cell proliferation and cell death assays**. Cell lines were reverse transfected with 10 nM siRNA or first seeded for attachment prior to drug treatments as described for the validation experiments.

For live-cell imaging and cell-growth assays, **c**ells were reverse transfected with siRNAs as detailed in the validation experiments in 96-well plates (Corning, CoStar®, Cat# CLS3595-50EA) in a 120 µl total volume and imaged using a 10X objective on an IncuCyte® S3 live-cell analysis system for upto 1 week. Confluence masks were defined and data was analysed using IncuCyte®Zoom software.

For 5′-ethynyl-deoxy-uridine (EdU)-incorporation cell cycle analysis, cells were pulse-labelled with 10 uM 5′-ethynyl-2′-deoxyuridine (EdU) (Jena Biosciences, Cat# CLK-N001) for 15 min, harvested, washed in 1% BSA-PBS, sequentially fixed in 4% paraformaldehyde and ice-cold 70% ethanol and stored at −20 °C. To stain mitotic cells, samples were washed in 1% BSA-PBS, permeabilized with 0.25% Triton X-100 in PBS for 20 min, blocked in 5% FCS in PBS, 30 min, incubated with pSer10 Histone H3-AlexaFlour® 647 conjugate (BioLegend, Cat# 650806), 1–2 h and washed in 1% BSA-PBS. EdU incorporation was detected by copper-coupled Click-iT chemistry (Cu-CCC) as previously described[51,52]. Cells were incubated with freshly prepared Click-iT reaction cocktail (50 mM Tris-HCl pH7.6, 150 mM NaCl, 4 mM CuSO$_4$, 1 µM Picolyl Azide 5/6-FAM (Jena Bioscience, Cat# CLK-1180), 2 mg ml$^{-1}$ Sodium-L-Ascorbate (SIGMA, #A4034) for 30 min at RT. Cells were washed and resuspended in 1% BSA-PBS with 0.5 µg ml$^{-1}$ DAPI for 30 min and measured on a BD LSR FORTESSA Flow Cytometer (BD Biosciences).

For caspase-3 activity assays, green fluorogenic caspase-3 substrate NucView™ 488 (Biotium, # 10402) was spiked at a 1:200 dilution as an apoptosis indicator at the start of the experiment and assayed using a 10X objective on IncuCyte® for upto 1 week.

Cell metabolic activity as a surrogate for cell viability was assayed after 4 h incubation using the CellTiter-Blue® reagent (Promega, Cat# G8081) as per manufacturer's recommendations, by measuring fluorescence on a Berthold™ plate reader at 560/590 nm.

Additionally, plates from validation experiments were washed in 1x Dulbecco's PBS (GIBCO, Cat# 14190094), fixed in 4% paraformaldehyde (VWR, Cat# VWRK4078-9005), permeabilised with 0.1% Triton X-100 (Sigma-Aldrich, Cat# 108603) in PBS, stained with Propidium Iodide/RNAse staining solution (BD Biosciences, Cat# 550825) and imaged with the 4X objective on the IncuCyte® to obtain whole-well images and absolute nuclear counts as a surrogate for cell numbers at the assay endpoint.

For PI exclusion cell death assays, $3–5 \times 10^4$ cells were seeded per well in 24-well plates, transfected with 10 nM siRNAs or treated with drugs as indicated. At the end of treatment, cells were harvested by pooling detached cells and gently disaggregating adherent cells by trypsinisation from each well, pelleted and washed once by centrifugation at 700×g for 5 min, resuspended in 250 ul of PBS. About 5 ul of 100 ug ml$^{-1}$ Propidium Iodide (SIGMA, Cat# P4170) solution in PBS was added to each sample, mixed well, incubated for 10 min, and measured on a BD Fortessa™ Flow Cytometer. Cells positive for PI were considered terminally dead, and PI-negative cells as live. BD FACS Diva and FlowJo V7.6.5 and V10.1.1 software packages were used for data analysis.

**Co-immunoprecipitation and mass spectrometry**. About 2 million HEK293T cells were seeded one day before transfection with 2–5 ug of the constructs pCMV3-RBBP9-FLAG (Sino Biological, # HG-16910-CF) and pIRES-Neo-FLAG-FANCC (cloned in-house, previously described). Plasmid DNA constructs were mixed with FuGENE reagent (Promega, Cat#E2311) in a 1:3 (µg: µl) ratio in low-serum OptiMEM medium (GIBCO, Cat# 31985047) as per the manufacturer's

instructions, allowed to complex for 15 min, gently layered drop-wise on cells and grown under standard cell culture conditions. After 48 h, transfected HEK293T cells were scraped in 1 ml of cold lysis buffer (1% Nonidet P-40 substitute; 10% glycerol; 50 mM MgCl2, 200 mM NaCl, Protease and phosphatase inhibitors) and cleared by centrifugation at 4 °C, 10,000 rpm. Cleared lysates were incubated with 40 µl anti-FLAG M2®™ beads (SIGMA, Cat# M8823) overnight at 4 °C while rotating. By placing Eppendorf tubes on a magnetic rack, beads were washed three times with lysis buffer and bound proteins were eluted using 100 µl of 150 ng µl$^{-1}$ of 3x FLAG-peptide (SIGMA, Cat# F4799), as per manufacturer's instructions and boiled in LDS sample buffer (NuPAGE, Cat# NP0007). IP samples were run along with 5% each of the cleared lysate as well as the IP supernatant, on pre-cast SDS-PAGE gels (Bio-Rad, 4–15%, Cat# 4561084DC) and blots developed after incubation with anti-FLAG antibody (SIGMA, Cat# F3165).

**Mass spectrometry-based proteomics using GeLC-MS/MS**. We applied our label-free GeLC-MS/MS-based proteomics workflow with an alternating study design that has been extensively bench-marked for reproducibility[53–55]

For LC-MS/MS sample preparation, equal protein (FLAG IP) eluates, were separated on pre-cast SDS-PAGE gels (Bio-Rad, 4–15%, Cat# 4561084DC). Gels were fixed in 50% ethanol/3% phosphoric acid solution and stained with Coomassie R-250. Gel lanes were cut into five bands and each band was cut into ~1 mm$^3$ cubes. Gel cubes were washed with 50 mM ammonium bicarbonate/50% acetonitrile and were transferred to a microcentrifuge tube, vortexed in 50 mM ammonium bicarbonate for 10 min and pelleted. The supernatant was removed, and the gel cubes were again vortexed in 50 mM ammonium bicarbonate/50% acetonitrile for 10 min. After pelleting and removal of the supernatant, this wash step was repeated. Subsequently, gel cubes were reduced in 50 mM ammonium bicarbonate supplemented with 10 mM DTT at 56 °C for 1 h, where after supernatant was removed. Gel cubes were alkylated in 50 mM ammonium bicarbonate supplemented with 50 mM iodoacetamide for 45 min at RT in the dark. Next, gel cubes were washed with 50 mM ammonium bicarbonate/50% acetonitrile, dried in a vacuum centrifuge at 50 °C and covered with trypsin solution (6.25 ng µl$^{-1}$ in 50 mM ammonium bicarbonate). Following rehydration with trypsin solution and removal of excess trypsin, gel cubes were covered with 50 mM ammonium bicarbonate and incubated overnight at 25 °C. Peptides were extracted from the gel cubes with 100 µl of 1% formic acid (once) and 100 µl of 5% formic acid/50% acetonitrile (twice). About 300 µl extracts were stored at −20 °C until use. Prior to LC-MS, the extracts were concentrated in a vacuum centrifuge at 50 °C, volumes were adjusted to 50 µl by adding 0.05% formic acid, filtered through a 0.45 um spin filter, and transferred to an LC autosampler vial.

For LC-MS/MS, peptides were separated by an Ultimate 3000 nanoLC-MS/MS system (Dionex LC-Packings, Amsterdam, Netherlands) equipped with a 50 cm × 75 µm ID Pepmap Acclaim C18 (1.9 µm, 120 Å)column (Thermo, Bremen, Germany). After injection, peptides were trapped at 3 µl min$^{-1}$ on a 10 mm × 75 µm ID Pepmap Acclaim C18 trap column in 0.1% formic acid. Peptides were separated at 300 nl min$^{-1}$ in a 10–40% gradient (buffer A: 0.1% formic acid (Fischer Scientific), buffer B: 80% ACN, 0.1% formic acid) in 90 min (120 min inject-to-inject). Eluting peptides were ionised at a potential of +2 kVa into a Q Exactive HF mass spectrometer (Thermo Fisher, Bremen, Germany). Intact masses were measured at resolution 70,000 (at m/z 200) in the orbitrap using an AGC target value of 3E6 charges. The top 15 peptide signals (charge-states 2+ and higher) were submitted to MS/MS in the HCD (higher-energy collision) cell (1.6 amu isolation width, 25% normalised collision energy). MS/MS spectra were acquired at resolution 17,500 (at m/z 200) in the orbitrap using an AGC target value of 1E6 charges, a maxIT of 60 ms, and an underfill ratio of 0.1%. Dynamic exclusion was applied with a repeat count of 1 and an exclusion time of 30 s.

For protein identification & label-free quantitation, MS/MS spectra were searched against a Swissprot Human FASTA file (release April 2020, 42347 entries, canonical and isoforms) using MaxQuant 1.6.10.43. Enzyme specificity was set to trypsin, and upto two missed cleavages were allowed. Cysteine carboxamidomethylation (Cys, +57.021464 Da) was treated as fixed modification and methionine oxidation (Met, +15.994915 Da) and N-terminal acetylation (N-terminal, +42.010565 Da) as variable modifications. Peptide precursor ions were searched with a maximum mass deviation of 4.5 ppm and fragment ions with a maximum mass deviation of 20 ppm. Peptide and protein identifications were filtered at an FDR of 1% using the decoy database strategy. The minimal peptide length was seven amino acids. Proteins that could not be differentiated based on MS/MS spectra alone were grouped into protein groups (default MaxQuant settings). Searches were performed with the label-free quantification option selected. Proteins were quantified by spectral counting, i.e., the number of identified MS/MS spectra for a given protein[56] combining the five fractions per sample. Raw counts were normalised on the sum of spectral counts for all identified proteins in a particular sample, relative to the average sample sum determined with all samples. To find statistically significant differences in normalised counts between sample groups, we applied the beta-binomial test (Pham, Piersma, Warmoes, & Jimenez, 2010), which takes into account within-sample and between-sample variation using an alpha level of 0.05.

**Drug treatments**. Emetine dihydrochloride (SIGMA, Cat# 316-42-7), a reported RBBP9 inhibitor, was used at a final concentration of 25 nM in an aqueous solution unless otherwise specified.

**Statistics and reproducibility**. Raw-fluorescence values from the arrayed genome-wide siRNA screens were read into R, $Log_2$-transformed, and the data normalised using Rscreenorm[57], to correct for plate effects and technical variations between screens. The computed lethality scores, herein referred to as model coefficient, ranged between 0 and 2 signifying lethal effects of the individual siRNA Smart-POOLs i.e., values near 0 = no effect, similar to non-targeting siRNA, values of 1 and above = strongly lethal, similar to positive controls PLK1 and/or UBB. For hit-calling based on differentially lethal effects between cell lines, or subgroups of cell lines, empirical-Bayes linear regression, as implemented by limma[58] was used. Computed p-values were corrected for multiple testing using Benjamini-Hochberg's FDR[59]. This part of the analysis was performed using R version 3.6.

For screen data: Gene–gene physical interactions for genes with siRNA screen lethality score ≥0.8 were extracted from the GeneMania database. Genes with established interactions were functionally annotated by the Reactom-pathway database with a cut-off of Benjamini FDR ≤0.05. Subsequently, the functional gene–gene interaction network was built by importing and combining the gene–gene interaction data, pathway information as well as the hits significance as log10-transformed FDR into the network visualisation tool Cytoscape.

For MS-data: High-confidence interactors were shortlisted based on statistics (p value ≤0.05) and further enriched based on higher peptide count representation in RBBP9-FLAG co-IPs compared to FANCC-FLAG co-IPs. The co-expression and physical interactions among these genes were obtained from GeneMania. Functional annotation was performed using the online GO analysis tool Toppgene. Benjamini FDR ≤0.05 was deemed significant. The data were subsequently imported into Cytoscape and the network was built.

Automated morphometric analysis of 3D spheroid cultures was performed using the MatLab-based application OrganoSeq as described by ref. [60]. Mean spheroid volumes were calculated and relative changes were expressed with respect to untreated controls for each cell line.

For all experiments other than primary screens, data are expressed as mean ($n = 3$) ± SD as indicated in the Figure legends. Non-parametric two-way ANOVA with Holm–Sidak correction for multiple comparison was performed to test statistical significance. $**p ≤ 0.01$, $***p ≤ 0.001$, $****p ≤ 0.0001$.

**Reporting summary**. Further information on research design is available in the Nature Portfolio Reporting Summary linked to this article.

## Data availability

Normalised data from all primary screens have been made available as excel files accompanying this article (Supplementary Data 1). The mass spectrometry proteomics data have been deposited to the ProteomeXchange Consortium via the PRIDE partner repository (http://www.ebi.ac.uk/pride) with the dataset identifier PXD026545 (see also Supplementary Data 2). Source data pertaining to Figs. 1, 2, 3 and 4 are furnished as Supplementary Data 4. Further information and requests for cell lines and disease models, expression constructs and raw data should be directed to and will be fulfilled by the Lead Contact, Josephine Dorsman (jc.dorsman@amsterdamumc.nl). This study did not generate any codes, novel or unique reagents.

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

## Acknowledgements
JdW is dearly missed and his contribution to this project and our team is fondly remembered. We thank former members of the Oncogenetics lab for their assistance with optimisation and performing siRNA screens. We thank Mohamad Amr Zaini for his helpful discussions and inputs to the coIP experiments. We acknowledge support from Ida van der Meulen, RNAi screening facility, Department of Medical Oncology, Amsterdam UMC for help with setting up the genome-wide screens. We thank Prof. Dr. Mario van der Stelt, Professor of Molecular Physiology, Leiden Institute of Chemistry, for fruitful discussions on the enzymology of RBBP9. This study was supported by a KWF research grant (VU 2013-5983) awarded to (late) Johan de Winter (JdW), RB, and VvB. An NWO middelgroot grant (91116017) awarded to CJ supported the mass spectrometry infrastructure of The Oncoproteomics Laboratory.

## Author contributions
C.S. and C.P. performed the screens. G.P. analysed the screen data, performed the majority of the experiments after the screens along with D.R. and drafted the manuscript. Y.d.J., K.R., M.R. and L.V. contributed to experimental designs and assisted with experiments. In addition, K.R. assisted with the data visualisation. SP performed the proteomics analysis. R.X.D.M. performed the statistical analysis of the screens. C.J. supervised the proteomics experiments. V.W.v.B. supervised the siRNA screens (RIFOL facility) and co-supervised C.S., C.P. and G.P. R.B., H.T.R. and R.M.F.W. co-supervised C.P. and G.P. J.C.D. supervised G.P., D.R., Y.d.J., K.R., M.R. and L.V., and co-supervised C.P., and co-wrote the manuscript with G.P. All authors contributed to discussions, read, reviewed and approved the manuscript.

## Competing interests
The authors declare no competing interests.
