## [Peer Review File · Communications Biology]

Reviewers' comments:

Reviewer #1 (Remarks to the Author):

In this study, authors identified RBBP9 as a candidate therapeutic target for head-and-neck squamous cell-carcinomas (HNSCC) of Fanconi anemia (FA) patients by siRNA screens to unveil genetic interactions synthetic-lethal with FA-pathway deficiency in FA-patient HNSCC cell lines. RBBP9-silencing resulted in profound lethality in FA-defective HNSCC cells, while lethality in FA-proficient HNSCC was less severe. Interestingly, RBBP9 silencing was significantly less toxic to FA-defective, patient-matched cancer associate fibroblasts and skin fibroblasts. Moreover, RBBP9-targeting drug Emetine kills FA-HNSCC cells, but not non-tumor cells. In summary, authors provide a new DNA cross-link-repair independent lead, RBBP9, for targeted treatment of FA-HNSCCs without systemic toxicity.

I feel that this paper contains interesting findings. However, authors examined the effects of RBBP9-targeting drug Emetine only in cell lines. To show the effects of Emetine for FA-HNSCCs, authors should perform additional experiments as described below. Therefore, the findings as reported do not advance our knowledge of the subject sufficiently to warrant publication in its form.

My comments are the following:

1. How about the difference of background of gene mutation between FA-HNSCCs and conventional HNSCCs?
2. Why is expression level of RBBP9 elevated in FA-HNSCCs? How about the mechanism?
3. In this study, authors only used FA-HNSCC cell lines. To know the importance of RBBP9 for therapeutic target, authors should check the expression of RBBP9 in clinical cases of FA-HNSCCs.
4. Authors showed the phenotype of RBBP9-silencing such as cell proliferation and sphere formation. How about the phenotype of RBBP9-overexpressing HNSCCs? Moreover, authors should describe the mechanism of RBBP9-silencing suppressed cell proliferation and sphere formation.
5. Authors showed the clear phenotype of Emetine treatment in FA-HNSCCs. To prove this phenotype, authors should check the effects of Emetine in mouse model (ex. orthotopic implantation of FN-HNSCC cells into mouse tongue).

Reviewer #2 (Remarks to the Author):

In this manuscript, Pai et al performed a genome-wide siRNA screens on two different Fanconi anemia (FA) pathway deficiency HNSCC cell lines (VU-SCC 1131 and VU-SCC 1604). In addition, the authors also performed a "corrected" FA-deficient cell line (VU-SCC 1131 + FACC) as a control (FA-proficient) for the screening results. By comparing the the overlapping synthetic lethality genes from the two FA-deficient lines and the differential-lethality between FA-deficient and FA-proficient cells, the authors identified 8 genes that are essential to FA-deficient HNSCC lines. Among these genes, PSMC1, PSMB2, and LAMTOR2 were known and previously described. RBBP9 is the gene that the authors identified to be essential in FA-deficient lines and put forward for validation. Using genetical and pharmacological validations, the authors showed that the low dose of the FDA-approved RBBP9-targeting drug Emetine kills FA-HNSCC lines. Importantly, the authors demonstrated that both RBBP9-silencing as well as Emetine spared non-tumour FA cells, in fibroblasts, stroma and human diploid CRISPR engineered cell lines. The results presented were supported by the data. Appropriate statistical cut-offs were used for filtering the gene lists. Overall, this is a well written manuscript.

However, there are some minor comments:

1. In the text, the authors mentioned VU-SCC 1365 line was also used as the primary screen, however, in Fig 1A and the results table (Supp Table 2), no data or mention of VU-SCC 1365. If VU-SCC 1365 was indeed used in the primary screen, that data need to be incorporated in Fig 2A (as well as provided in Supp Table 2) to show in the Venn diagram. This needs to clarify by the authors.
2. The FA-deficient lines were all TP53 mutant lines, do the authors have other genomics / mutational data for these lines?

3. Fig 1B, the authors should include labels for y-axis and x-axis. The criteria for FDR should be $> -\log_{10}(11.5)$ or $FDR < 10^{-11.5}$ to be correct. Similarly, in Fig 2A, the FDR thresholds should be written as $FDR > -\log_{10}(10)$ or $FDR < 10^{-10}$.

4. The interaction networks presented in Figs 1 and 2 require some descriptions in Methods. Currently, it is unclear how the authors generated these networks.

5. The authors made the claim that "This study provides the first minable genome-wide analyses of vulnerabilities to address treatment challenges in FA-HNSCC." However, the "minable" is the "excel file" provided as Suppl Table 2? Or do the authors have a web portal for user to "mine" the results?

Rebuttal Letter – Pai M. G. *et al.*

Reviewers' comments:

Reviewer #1 (Remarks to the Author):

In this study, authors identified RBBP9 as a candidate therapeutic target for head-and-neck squamous cell-carcinomas (HNSCC) of Fanconi anemia (FA) patients by siRNA screens to unveil genetic interactions synthetic-lethal with FA-pathway deficiency in FA-patient HNSCC cell lines. RBBP9-silencing resulted in profound lethality in FA-defective HNSCC cells, while lethality in FA-proficient HNSCC was less severe. Interestingly, RBBP9 silencing was significantly less toxic to FA-defective, patient-matched cancer associate fibroblasts and skin fibroblasts. Moreover, RBBP9-targeting drug Emetine kills FA-HNSCC cells, but not non-tumor cells. In summary, authors provide a new DNA cross-link-repair independent lead, RBBP9, for targeted treatment of FA-HNSCCs without systemic toxicity.

I feel that this paper contains interesting findings. However, authors examined the effects of RBBP9-targeting drug Emetine only in cell lines. To show the effects of Emetine for FA-HNSCCs, authors should perform additional experiments as described below. Therefore, the findings as reported do not advance our knowledge of the subject sufficiently to warrant publication in its form.

1. How about the difference of background of gene mutation between FA-HNSCCs and conventional HNSCCs?

The FA-HNSCC cell lines as well as patient-matched cancer-associated fibroblasts and/or skin fibroblasts, as indicated in this manuscript, have also been reported in a recent manuscript from our lab. Please see Roohollahi *et al.*, Scientific reports 2022 PMID: 34997070 (BIRC2–BIRC3 amplification: a potentially druggable feature of a subset of head and neck cancers in patients with Fanconi anemia | Scientific Reports (nature.com)) for detailed genomic and transcriptomic characterisation of these head-and-neck cell lines, as well as a comparison with sporadic HNSCC.

2. Why is expression level of RBBP9 elevated in FA-HNSCCs? How about the mechanism?

We do not claim by any means that the expression levels of RBBP9 are elevated in FA-HNSCC. In fact, we are aware of the fact that the median expression levels are even modestly higher in sporadic HNSCC cell lines, however, with no overall significant difference. We only report an increased dependency of FA-HNSCC on RBBP9. Relative expression levels of RBBP9 in FA-HNSCC and sporadic HNSCC cell lines are shown in Figure 1 below (genome-wide data available from Roohollahi *et al.*, Scientific reports 2022 PMID: 34997070).

Figure 1. RNA expression data. RBBP9 expression levels across Fanconi Anaemia (FA) / Sporadic (SP) tumours (HNSCC) and their respective fibroblasts (CAF). Log2FC (FA-T vs. SP-T) = 0.57, FDR=0.67

3. In this study, authors only used FA-HNSCC cell lines. To know the importance of RBBP9 for therapeutic target, authors should check the expression of RBBP9 in clinical cases of FA-HNSCCs.

FA is a rare disease; head-and-neck cancer FA patients constitute a very small patient cohort worldwide. Therefore access to clinical specimens is extremely limited. To the best of our knowledge the limited protein studies have been focused on our rare cell lines. Publicly available databases indicate that RBBP9 is ubiquitously expressed and displays no tissue specificity. Here, we propose that RBBP9 function, rather than its expression levels, is crucial for FA-HNSCC tumour lines investigated in this study. Relative expression levels of RBBP9 in FA-HNSCC and sporadic HNSCC cell lines available have been presented above (see remark 2).

4. Authors showed the phenotype of RBBP9-silencing such as cell proliferation and sphere formation. How about the phenotype of RBBP9-overexpressing HNSCCs? Moreover, authors should describe the mechanism of RBBP9-silencing suppressed cell proliferation and sphere formation.

We wish to reiterate here that the expression levels of RBBP9 are not higher in the FA-HNSCC cell lines per se.

As suggested, we have now, nevertheless, performed these requested experiments by overexpressing the orange fluorescent-tagged (RBBP9-OFP) protein in the panel of eight FA-HNSCC and ARPE-19 hTERT cell lines used in this study. Making use of the NucView-488 green-fluorogenic Caspase-3 cleavage substrate, we have quantified the orange-green double positive population. In general, within an acute

period of 72h post-transfection, modest toxicity of RBBP9-OFP overexpression could be observed (\pm 20% apoptotic cells) as compared to the empty vector control transfection in this time period (n=3 independent experiments, please see data below). We observed no other discernible phenotype within this time-frame. Interestingly, attempts to generate stable cell lines expressing RBBP9-OFP or RBBP9-FLAG were not successful in FA-HNSCC lines, but clearly possible in the untransformed ARPE-19 hTERT lines (data not shown). In total these data suggested that sustained abundance of RBBP9 may have deleterious consequences in FA-HNSCC cell lines too. Since this was not the topic of our manuscript, i.e. identification of targets whose KD has a deleterious effect, we decided not to pursue this topic further. We have added a short remark though in the text that overexpression of RBBP9 does not seem to be well-tolerated (lines 169 – 171).

Figure 2: RBBP9-overexpression experiments. Representative flow cytometry histograms (A) indicating transfection efficiency of pcDNA EV-mCherry and pcDNA-RBBP9-OFP respectively in the FA-HNSCC and ARPE-

19 hTERT cell lines, quantified in B (left panel, n=3). The Caspase-3 cleavage-positive apoptotic fraction with RBBP9 overexpression present after 72 hours (Red-green double positive, single cells) is presented in B, right.

As presented in Fig. 3d, RBBP9-silencing by siRNA resulted in Caspase-3-mediated apoptosis in monolayer cultures. We have addressed now the mode of cell death after RBBP9-inhibition by Emetine in spheroid assays. Caspase-3 mediated apoptosis appears to take precedence over other cell death mechanisms as this could be detected as early as 24h after Emetine treatment (Please see Suppl. Fig. 4f, description of the results in lines 194 - 196).

5. Authors showed the clear phenotype of Emetine treatment in FA-HNSCCs. To prove this phenotype, authors should check the effects of Emetine in mouse model (ex. orthotopic implantation of FN-HNSCC cells into mouse tongue).

This is indeed of great interest to our lab. In fact, such pre-clinical animal studies are being considered for a follow-up study, as drug-repurposing for cancer is clearly a good strategy especially in rare diseases such as FA, where clinical trials can be limited by cohort size. However, these experiments are beyond the scope of the content presented in the current manuscript.

Reviewer #2 (Remarks to the Author):

In this manuscript, Pai et al performed a genome-wide siRNA screens on two different Fanconi anemia (FA) pathway deficiency HNSCC cell lines (VU-SCC 1131 and VU-SCC 1604). In addition, the authors also performed a "corrected" FA-deficient cell line (VU-SCC 1131 + FACC) as a control (FA-proficient) for the screening results. By comparing the the overlapping synthetic lethality genes from the two FA-deficient lines and the differential-lethality between FA-deficient and FA-proficient cells, the authors identified 8 genes that are essential to FA-deficient HNSCC lines. Among these genes, PSMC1, PSMB2, and LAMTOR2 were known and previously described. RBBP9 is the gene that the authors identified to be essential in FA-deficient lines and put forward for validation. Using genetical and pharmacological validations, the authors showed that the low dose of the FDA-approved RBBP9-targeting drug Emtine kills FA-HNSCC lines. Importantly, the authors demonstrated that both RBBP9-silencing as well as Emetine spared non-tumour FA cells, in fibroblasts, stroma and human diploid CRISPR engineered cell lines. The results presented were supported by the data. Appropriate statistical cut-offs were used for filtering the gene lists. Overall, this is a well written manuscript.

However, there are some minor comments:

1. In the text, the authors mentioned VU-SCC 1365 line was also used as the primary screen, however, in Fig 1A and the results table (Supp Table 2), no data or mention of VU-SCC 1365. If VU-SCC 1365 was indeed used in the primary screen, that data need to be incorporated in Fig 2A (as well as provided in Supp Table 2) to show in the Venn diagram. This needs to clarify by the authors.

We thank the reviewer for pointing out that the VU-SCC 1365 cell line has been mentioned in the methods section. The authors wish to clarify that the VU-SCC 1365 cell was in fact used for the primary screen. Unfortunately, technical issues with all replicates of the screen in this particular cell line resulted in data quality that was unsuitable for our purpose of hit identification. Cognizant of the fact that FANCA represents the major complementation group and the most frequently mutated FA gene, the authors have chosen to call hits from screens performed in VU-SCC 1131, VU-SCC 1131+FANCC and VU-SCC 1604, while using VU-SCC 1365 cell line in all subsequent validation and mechanistic studies. We have now amended the methods text accordingly (lines 252 – 254).

2. The FA-deficient lines were all TP53 mutant lines, do the authors have other genomics / mutational data for these lines?

The authors acknowledge that the patient-derived FA-HNSCC lines are all TP53-mutant, in contrast to their fibroblast counterparts and the CRISPR-engineered ARPE-19 hTERT FA-model cell lines. The FA-HNSCC cell lines as well as patient-matched cancer-associated fibroblasts and/or skin fibroblasts, as indicated in this manuscript, have also been reported in a recent manuscript from our lab. Please see Roohollahi *et. al.*, Scientific reports 2022 PMID: 34997070 (BIRC2–BIRC3 amplification: a potentially druggable feature of a subset of head and neck cancers in patients with Fanconi anemia | Scientific Reports (nature.com)) for detailed genomic and transcriptomic characterisation of these cell lines.

3. Fig 1B, the authors should include labels for y-axis and x-axis. The criteria for FDR should be $> -\log_{10}(11.5)$ or $FDR < 10^{-11.5}$ to be correct. Similarly, in Fig 2A, the FDR thresholds should be written as $FDR > -\log_{10}(10)$ or $FDR < 10^{-10}$.

We thank the reviewer for pointing out this inadvertent error. The corrections are now incorporated.

4. The interaction networks presented in Figs 1 and 2 require some descriptions in Methods. Currently, it is unclear how the authors generated these networks.

We wish to point out that Figure 1 (and not Figure 2) has a network, as do Suppl. Figures 1a and 1b. We have included a description of how the networks were generated in the methods section (lines 497, 507 – 518) in addition to the figure legends.

5. The authors made the claim that "This study provides the first minable genome-wide analyses of vulnerabilities to address treatment challenges in FA-HNSCC." However, the "minable" is the "excel file" provided as Suppl Table 2? Or do the authors have a web portal for user to "mine" the results?

We refer to the normalised lethality scores provided as an excel sheet. We apologise if the reviewer(s) were misled. We have now rephrased the text and amended "minable" to "publicly available" (line 236).

REVIEWERS' COMMENTS:

Reviewer #1 (Remarks to the Author):

I feel that the authors satisfactory responded to the comments from two reviewers.

Reviewer #2 (Remarks to the Author):

In this revised manuscript, the authors have satisfactory addressed my previous comments. The findings from this study will be impactful to the research community.